# Analysis of the Prognostic Potential of Schlafen 11, Programmed Death Ligand 1, and Redox Status in Colorectal Cancer Patients

**DOI:** 10.3390/ijms242015083

**Published:** 2023-10-11

**Authors:** Marko Miladinov, Jovana Rosic, Katarina Eric, Azra Guzonjic, Jelenko Jelenkovic, Natasa Bogavac-Stanojevic, Ivan Dimitrijevic, Jelena Kotur-Stevuljevic, Goran Barisic

**Affiliations:** 1Clinic for Digestive Surgery—First Surgical Clinic, University Clinical Center of Serbia, 11000 Belgrade, Serbia; marko.miladinov90@gmail.com (M.M.); jelenkojelenkovic1@gmail.com (J.J.); ivanclean@gmail.com (I.D.); 2Faculty of Medicine, University of Belgrade, 11000 Belgrade, Serbia; jovana.rosic@med.bg.ac.rs; 3Department of Pathology, University Clinical Center of Serbia, 11000 Belgrade, Serbia; ketrineric@gmail.com; 4Faculty of Pharmacy, University of Belgrade, 11000 Belgrade, Serbia; azra.guzonjic@pharmacy.bg.ac.rs (A.G.); natasa.bogavac@pharmacy.bg.ac.rs (N.B.-S.); jelena.kotur@pharmacy.bg.ac.rs (J.K.-S.)

**Keywords:** colorectal cancer, redox status, oxidative stress, SLFN11, PD-L1, biomarkers

## Abstract

The Schlafen 11 (SLFN11) protein has recently emerged as pivotal in DNA damage conditions, with predictive potential for tumor response to cytotoxic chemotherapies. Recent discoveries also showed that the programmed death ligand 1 (PD-L1) protein can be found on malignant cells, providing an immune evasion mechanism exploited by different tumors. Additionally, excessive generation of free radicals, redox imbalance, and consequential DNA damage can affect intestinal cell homeostasis and lead to neoplastic transformation. Therefore, our study aimed to investigate the significance of SLFN11 and PD-L1 proteins and redox status parameters as prognostic biomarkers in CRC patients. This study included a total of 155 CRC patients. SLFN11 and PD-L1 serum levels were measured with ELISA and evaluated based on redox status parameters, sociodemographic and clinical characteristics, and survival. The following redox status parameters were investigated: spectrophotometrically measured superoxide dismutase (SOD), sulfhydryl (SH) groups, advanced oxidation protein products (AOPP), malondialdehyde (MDA), pro-oxidant–antioxidant balance (PAB), and superoxide anion (O_2_^•–^). The prooxidative score, antioxidative score, and OXY-SCORE were also calculated. The results showed significantly shorter survival in patients with higher OXY-SCOREs and higher levels of serum SLFN11, while only histopathology-analysis-related factors showed significant prognostic value. OXY-SCORE and SLFN11 levels may harbor prognostic potential in CRC patients.

## 1. Introduction

Colorectal cancer (CRC) represents a significant global public health problem. With over 1.9 million affected people worldwide in 2020, it is the third most common type of cancer, after lung and breast cancers. According to World Health Organization projections, the number of newly diagnosed CRC cases will increase to 3.1 million annually by 2040 [1], with an essential rise seen in those under 50 years of age [2]. Additionally, with over 930,000 deaths yearly, CRC ranks second in mortality rate following lung cancer [1]. Modern treatment of CRC involves a combination of different therapeutic modalities, such as surgical treatment, chemotherapy, radiotherapy, and immunotherapy. However, local recurrence and distant metastases occur in almost half of the patients, drastically shortening survival [3,4]. Therefore, there is an urgent need for a better understanding of the molecular mechanisms underlying CRC, the contribution of various risk factors to disease outcomes, and novel sensitive and specific prognostic biomarkers.

CRC is a multifactorial disease arising from the complex interplay between genetic and environmental factors. In 70% of all cases, it occurs as a sporadic disease, while there is a familial risk or hereditary component in the remaining 30% [5]. Risk factors such as pollution, stress, lack of physical activity, obesity, a diet rich in fats and refined sugars, smoking, and alcohol consumption are considered to have a significant impact [6].

It was demonstrated that many environmental risk factors lead to oxidative stress, which can initiate carcinogenesis by damaging DNA molecules [7]. Oxidative stress is caused by an imbalance between the production of prooxidants, so-called reactive oxygen species (ROS), and the antioxidant protection mechanisms. It is quantified using a set of parameters, namely prooxidants and products of their action—superoxide anion (O_2_^•–^), oxidatively modified lipids (malondialdehyde, MDA), oxidatively modified proteins (advanced oxidation protein products, AOPP), combined indicators of oxidative stress (total oxidant status and prooxidant–antioxidant balance (PAB))—as well as a set of parameters of antioxidant protection, namely non-enzymatic parameters (reduced glutathione and total sulfhydryl groups (tSHG)) and enzymatic parameters (total activity of the superoxide-dismutase (SOD) enzyme). Under physiological conditions, ROS is predominantly generated in the respiratory chain in mitochondria and acts as signal molecules. In oxidative stress conditions, increased production of these molecules occurs, leading to manifestation of their harmful effects.

ROS oxidize lipids, proteins, and DNA, thus compromising the structure and function of these molecules, resulting in structural damage to cells and alterations to various signaling pathways, which stimulates inflammation and the process of carcinogenesis [8]. Damage to the DNA molecule by breaking one or both strands leads to disturbances in the cell cycle, which further leads to mutation and carcinogenesis [9]. Numerous studies investigated the influence of free oxygen radicals on the development of CRC over the last 40 years. Many genes involved in CRC initiation and progression, such as *APC*, *p53*, *KRAS*, and *BRAF*, are susceptible to ROS-induced mutations [10]. ROS can also modify various transcription factors and upregulate the epithelial–mesenchymal transition (EMT) process, the mechanism crucial for metastatic disease development [11]. On the other hand, ROS can trigger the programmed cell death process, which can be utilized in anticancer therapy strategies. However, by gaining a response against ROS action, tumor cells can acquire resistance to anticancer remedies [12]. Despite a large number of studies concerning this topic, ROS-mediated mechanisms of action in the development and progression of CRC still need to be elucidated [12].

*SLFN11*, a Schlafen (SLFN) gene family member, codes for a homonymous protein essential in DNA molecule damage conditions. This putative DNA/RNA helicase acts as an S-phase checkpoint and induces an irreversible replication block by binding to the replication fork in conditions of replication stress [13,14]. This process involves chromatin opening nearby replication initiation sites, thus inducing replication blockage and ultimately leading to cell death [13,15]. This molecular mechanism highlighted SLFN11 as a promising predictive biomarker for response to cytotoxic chemotherapies, particularly DNA-damaging agents (DDAs) such as topoisomerase I and II, DNA synthesis inhibitors, cross-linkers, and alkylating agents [14,15,16,17]. SLFN11 is recruited directly to the stalled replication fork in response to replication stress induced by DDAs, and its expression levels exhibit a strong positive correlation with tumor sensitivity to DDAs [18]. Conversely, the inactivation of *SLFN11* by hypermethylation is the mechanism of epigenetic resistance to these anticancer drugs [13]. Previous studies demonstrated the predictive value of *SLFN11* in many cancers, including small-cell lung [17], ovarian [19], and colorectal [20] cancers. *SLFN11* expression is usually quantified using immunohistochemistry (IHC) assays, RNA sequencing, and methylome analyses [21], but studies measuring its concentrations in circulation are still lacking.

Programmed death ligand 1 (PD-L1) is a transmembrane glycoprotein that plays a vital role in maintaining immune tolerance by binding to its receptor PD-1 and inhibiting T-lymphocyte activity [22]. It is commonly expressed in lymphocytes and antigen-presenting cells and tissues like the placenta, testes, and eyes. PD-L1 protein can also be found on malignant cells, providing an immune evasion mechanism exploited by different tumor types [23], including melanoma, non-small-cell lung cancer, ovarian and breast, and gastrointestinal malignancies. In CRC, determining the serum levels of PD-L1 may have prognostic significance since previous studies showed a negative correlation of PD-L1 serum levels in CRC patients with overall survival [24,25].

Oxidative stress is central to our research since it has been elaborately studied and found significant in carcinogenesis and chronic inflammation. SLFN11 and PD-L1 have been chosen since they have been in the limelight of colorectal cancer research in recent years, with an essential role as immunological checkpoints and potential therapeutical targets as well as biomarkers for response and prediction of therapy outcomes. Although the mechanisms underlying their involvement in CRC are not fully elucidated, we wanted to examine their potential as circulating biomarkers solely and combined for the prognosis of disease outcome [13,14,16,22,23].

Our study aimed to investigate the significance of redox status parameters, and the SLFN11 and PD-L1 proteins, as well as their combined effect as prognostic biomarkers in patients with colorectal cancer.

## 2. Results

### 2.1. The Socio-Demographic and Clinical Characteristics of CRC Patients

The socio-demographic and clinicopathological characteristics of patients with CRC included in this study are summarized in Table 1.

### 2.2. Comparison of Redox Status Parameters between CRC Patients and the Healthy Control Group

The redox status parameters (PAB, AOPP, MDA, O_2_^•–^, SOD, and SHG) measured in the serum of 155 CRC patients and 60 healthy controls are shown in Table 2. The concentrations of AOPP and superoxide anion are significantly higher in the serum of CRC patients relative to the control group (*p* < 0.001), as well as the concentration of MDA (*p* = 0.002). On the other hand, the concentrations of antioxidants SOD and SHG are significantly lower in the serum of CRC patients (*p* < 0.001 and *p* = 0.002, respectively). There is no significant difference in the PAB concentrations in both groups.

### 2.3. Calculated Prooxidative Score, Antioxidative Score, and Summary OXY-SCORE in CRC Patients and the Healthy Control Group

The prooxidative score, antioxidative score, and summary OXY-SCORE calculated for 155 CRC patients and 60 healthy controls in order to obtain comprehensive insight into their redox statuses are presented in Figure 1 as box plots. In CRC patients, the antioxidative score is significantly lower relative to the control group (*p* < 0.001). Contrarily, the prooxidative score and summary OXY-SCORE are significantly higher in CRC patients compared with those in the control group (*p* < 0.001).

### 2.4. The Influence of Lifestyle and Environmental Factors on Measured and Calculated Parameters of Redox Status in CRC Patients

According to the results of our study, smoking caused significant redox status disturbance, measured through the AOPP and overall prooxidative score increase (*p* < 0.05 for both) (Figure 2A). Conversely, lower oxidative stress was confirmed in patients with active lifestyles based on the lower superoxide anion concentrations and prooxidative score values (*p* < 0.05 for both) (Figure 2B). Increased exposure to radiation led to a significant increase in AOPP concentrations and lowering of SOD activity (*p* < 0.05 for both) (Figure 2C).

### 2.5. Kaplan–Meier Analysis of the Survival of CRC Patients According to OXY-SCORE Value Risk

A Kaplan–Meier analysis of the overall survival of 135 CRC patients after a 3-year following period is shown in Figure 3. The patients with OXY-SCOREs above the 75th percentile (34 patients) had a significantly shorter survival period compared with those with OXY-SCOREs below the 75th percentile (101 patients). The median survival of patients with lower and higher OXY-SCOREs was 34 ± 7 months and 31 ± 10 months, respectively.

### 2.6. The Relationship between Measured and Calculated Redox Status Parameters and SLFN11 and PD-L1 Protein Serum Concentrations, and One- and Three-Year Survival of CRC Patients

The relationship between SLFN11 and PD-L1 protein serum concentrations and one- and three-year survival of CRC patients is shown in Figure 4 as box plots. The SLFN11 protein levels are significantly higher in the serum of patients who died during the first year of surveillance (*p* = 0.035). On the other hand, the measured and calculated redox status parameters (PAB, AOPP, MDA, O_2_^•–^, SOD, SHG, prooxidative score, antioxidative score, and OXY-SCORE) and PD-L1 protein levels in the serum did not differ significantly among living patients and those who died in the first year of surveillance. The latter had higher SOD activities, but they did not reach statistical significance.

The patients who died during the complete study follow-up had significantly higher SLFN11 protein levels (*p* ≤ 0.05), while the abovementioned redox status parameters and PD-L1 protein levels in the serum did not differ significantly.

### 2.7. Principal Component Analysis (PCA) of All Measured and Calculated Variables in CRC Patients and Binary Logistic Regression Analysis of PCA-Extracted Factors for One-Year Mortality Prediction

A factorial analysis (principal component analysis (PCA)) was performed to reduce the number of variables into a smaller number of factors formed from the parameters with the same level of variability. An analysis of sampling adequacy showed satisfying results (Kaiser–Meier–Olkin estimate was 0.619), and Bartlett’s test confirmed sphericity existence (*p* < 0.001). The analysis extracted three significant factors explaining 69% of the total variability. The factors’ variables with loadings and percentages of total variability per factor are presented in Table 3.

Factors’ related scores from the factorial analysis were used for subsequent binary logistic regression analysis for prediction of mortality in the first study year. The results of an univariant and multivariant binary logistic regression analysis are presented in Table 4. According to this analysis, histopathology-analysis-related factors are significant predictors of mortality during the first year after diagnosis establishment in the univariant analysis as well as in the multivariant analysis (*p* = 0.010 and *p* = 0.009, respectively). Two other factors—the immuno-modulatory-oxidative stress- and redox-status-related factors—did not reach statistical significance for the one-year mortality prediction.

## 3. Discussion

The number of CRC deaths worldwide is predicted to increase from the current 900,000 to close to 1.6 million annually by 2040, with an almost two-fold increase in the number of new cases, which will continue to impose a significant economic burden [1,26]. Therefore, it is necessary to uncover new prognostic and predictive biomarkers in addition to new therapeutic modalities. Recently, different biomarker-discovering strategies have been utilized, and many potential genetic and epigenetic biomarkers and their signatures with prognostic and predictive potentials in CRC have been discovered [27]. Validation of emerging biomarkers would allow for the personalization of therapy and improvement in treatment outcomes.

Previous studies revealed that oxidative stress, implicated in various diseases, may be an important progenitor in carcinogenesis, including CRC. Excessive generation of free radicals, redox imbalance, and consequential DNA damage can affect intestinal cell homeostasis and lead to neoplastic transformation. Both cancer-suppressing and cancer-promoting roles of ROS have been previously indicated, and this dichotomy is presumed to be level-dependent [28]. ROS can stimulate cell proliferation, apoptosis and anoikis avoidance, tissue invasion and angiogenesis, and the EMT [28]. Contrarily, ROS can also be involved in specific antitumoral responses such as T lymphocytes and natural killer cell activations [29]. High levels of oxidative stress can be present in cancer cells due to malignant transformations, including the changes in the tumor microenvironment [30]. In the present study, we observed significantly higher ROS concentrations in CRC patients compared with the control group, as expected, since it has already been demonstrated in CRC [31] and other tumor types [32,33]. Similarly, Oberly et al. found a reduced concentration of antioxidants in clear cell renal carcinoma cells [34], which is also consistent with our findings.

The overproduction of ROS and oxidative stress can be triggered by different lifestyle and environmental factors, including smoking, alcohol consumption, diet, lack of physical activity, infection, and radiation exposure, all factors considered to impact CRC development [35]. The influence of smoking on lung cancer development was previously undoubtedly proven, but a relation to CRC development is not straightforward. In their meta-analysis, Liang et al. confirmed the association between smoking and colorectal cancer development [36]. Our study showed that smoking is associated with a significant increase in AOPP concentration and the prooxidative score, concordant with previous findings supporting the prooxidative effect of tobacco smoke and its ingredients [37]. The prooxidative score was only significant among calculated parameters, while AOPP was only significant among measured parameters influenced by smoking, so only these two results were presented. A sedentary lifestyle, often linked with obesity, can increase the risk of CRC development. Conversely, moderate physical activity increases metabolism and decreases blood pressure, providing protective effects and reducing the risk of CRC development [38,39]. In our study, CRC patients with moderate physical activity had lower levels of oxidative stress (decreased superoxide anion concentrations and prooxidative score values), which is in line with previous investigations [40]. Cell exposition to ionizing radiation causes a complex cascade of molecular reactions leading to DNA damage and cell death [37]. One of the mechanisms is excessive ROS production, an effect that is exerted in a dose-dependent manner [41]. That is concordant with our results which show that increasing radiation exposition leads to increasing AOPP levels and decreasing SOD levels.

Different overview scores can be calculated from concentrations of the redox status parameters to assess the patient’s overall redox status. In 2006, Veglia et al. proposed the OXY-SCORE as a comprehensive index of oxidative stress status. It considers risk factors (prooxidants), protective factors (antioxidants), and markers of tissue damage to obtain the number representing a comprehensive index of risk [42]. OXY-SCORE is utilized for a better understanding of redox processes in the body. Its high and positive values indicate the predominance of the prooxidant processes. In contrast, its low or negative values indicate the predominance of antioxidant processes, i.e., that the organism successfully overcomes oxidative stress in physiological and pathological processes. This score was previously used for oxidative stress assessment in patients with cardiovascular disease [43,44]. In our study, OXY-SCORE was significantly higher in CRC patients than in healthy controls. This result is expected considering that OXY-SCORE is elevated under oxidative stress conditions. An analysis of the overall survival of patients with CRC revealed that it is longer in patients with lower OXY-SCORE, which makes this score a candidate prognostic biomarker. However, due to the uneven distribution of patients above and below the 75th percentile as a consequence of limited patient numbers, future studies should confirm our results in a larger patient cohort. To our best knowledge, this is the first study assessing the utility of OXY-SCORE in CRC patients as a prognostic biomarker. In addition to OXY-SCORE, we investigated other candidate parameters with prognostic potential for CRC. 

The SLFN11 protein recently emerged as pivotal in DNA damage conditions, with predictive potential for tumor response to cytotoxic chemotherapies, particularly DDAs. Our study showed higher serum SLFN11 levels in patients with shorter one-year and three-year overall survival. Conversely, in an immunohistochemical analysis by Deng et al., patients with higher SLFN11 expression had significantly longer three-year survival [45]. A possible explanation for this discrepancy could be that the tumor burden is higher in people with shorter survival, higher concentrations of SLFN11 are released into the blood, and the measured concentrations in the serum are higher. Further studies are needed to confirm this hypothesis. Zoppoli et al. first identified a strong positive correlation between the *SLFN11* gene expression and the cytotoxicity profile of DNA-targeting anticancer drugs [46]. Since then, SLFN11 has been examined as a potential predictive biomarker for DDA response in many preclinical studies [16,47]. Also, the potential of SLFN11 as a prognostic biomarker has been confirmed in hepatocellular [48], gastric [49], esophageal [50], and bladder [51] cancer, in the majority of which SLFN11 expression was examined by immunohistochemistry. To our best knowledge, this is the first study where SLFN11 concentrations were measured in serum using the ELISA method.

Recent discoveries showed that the PD-L1 protein can be found on malignant cells, providing an immune evasion mechanism exploited by different tumors. In CRC, tumoral expression of the membrane-bound receptor form of PD-L1 (mPD-L1) is a rare characteristic strongly associated with PD-1-positive lymphocytic infiltrates and deficiency in mismatch-repair systems, which are markers predicting high immunogenicity and responsiveness to anti-PD-1/PD-L1 therapies [52]. However, a previous study by Dank et al. showed that high levels of the PD-L1 soluble form (sPD-L1) in plasma were significantly associated with increased tumor burden and shorter disease-specific survival and progression-free survival in metastatic CRC patients [25]. Similarly, Omura et al. confirmed the prognostic potential of both sPD-L1 and mPD-L1 in stage I–III CRC patients, where elevated preoperative sPD-L1 levels were significantly correlated with lymphatic invasion, and both high tumoral mPD-L1 and elevated preoperative sPD-L1 were significantly associated with shorter overall survival and disease-free survival [24]. Contrarily, our study did not find associations between PD-L1 protein serum concentrations and one- and three-year survival. Increased PD-L1 concentrations were observed in the serum of patients with shorter survival, although this trend did not achieve statistical significance.

Our study showed that only pathophysiology-related factors significantly predict mortality during the first year after diagnosis in univariant and multivariate binary logistic regression analyses of PCA-extracted factors. This result is somewhat expected because of the coherence between pathohistological grade and patients’ status and disease severity [53].

## 4. Materials and Methods

### 4.1. Subjects

This prospective study was performed at the Clinic for Digestive Surgery—First Surgical Clinic University Clinical Center of Serbia and Faculty of Pharmacy, University of Belgrade. The patients’ blood samples were collected from January 2019 to January 2020, while the surveillance of the patients was conducted from January 2020 to January 2023. The study was conducted in accordance with the Declaration of Helsinki and approved by the Ethics Committee of the University Clinical Center of Serbia (number 14/4 from 25 January 2019). Before inclusion, written informed consent was obtained from all study participants.

We included a total of 155 patients diagnosed with colorectal cancer in the specified period. The inclusion criterion was histopathologically verified adenocarcinoma in any part of the large intestine. The exclusion criteria were previous chemo/chemoradiotherapy, metastatic disease, synchronous malignancies, severe comorbidities (ASA score of more than 3), and reluctance to participate. The control group consisted of 60 subjects who did not have the investigated disease or any other chronic non-infectious or infectious disease. 

After hospitalization, we obtained a social-epidemiological survey from all study participants, which included the following: 1. general information (gender, age, ethnicity, and level of education); 2. physical characteristics (body height and body mass); 3. personal history (comorbidities); 4. family history; 5. tobacco consumption; 6. alcohol consumption; and 7. occupation and physical activity.

The fasting blood samples were obtained safely from patients with minimal loss and contamination risk. Two blood samples were obtained using Vacutainer tubes (BD New Jersey, USA) with EDTA as an anticoagulant (10 mL and 3 mL vacutainers). A total of 3 mL was stored as whole blood, while 10 mL was divided into red blood cells, buffy coat, and plasma. The blood samples were centrifuged for 10 min at 2000× *g* with acceleration and deceleration scores of 4. The blood samples were stored at −80 °C until further analyses.

The standard histopathological analysis was performed using the Eight Edition of the American Joint Committee on Cancer (AJCC) T-N-M (TNM) staging system, as well as Dukes and Astler-Coller classifications [54]. In addition, lymphovascular and perineural invasion and resection margins status were determined.

Patients underwent standard diagnostic and therapeutic protocols during hospitalization, and all information was retrieved from the medical records. In addition, follow-ups of operated patients were carried out, and data on survival, disease status (remission, progression), and post-operative therapy were collected by telephone survey after one and three years for 135 available patients. The laboratory and molecular analyses were conducted at the Faculty of Pharmacy University of Belgrade.

### 4.2. Analyses of Redox Status Parameters

We used spectrophotometry to assess the following prooxidant and antioxidant species and to evaluate oxidative stress status in the CRC tissue and plasma/serum of 155 included patients: as markers of oxidative stress, we measured the concentrations of PAB (U/L), AOPP (μmol/L), MDA (μmol/L), and O_2_^•–^ levels (μmol NBT/min/L), while as markers of antioxidant protection, we measured the concentrations of SOD (U/L) and tSHG (mmol/L) content. All spectrophotometric measurements which do not include precipitation and centrifugation were implemented in an ILAB 300 Plus analyzer (Instrumentation Laboratory, Milan, Italy) [55]. All reagents were purchased from Sigma-Aldrich Chemie (Munich, Germany).

#### 4.2.1. PAB Concentration Determination

PAB concentrations were determined with a modified PAB test using 0.6%, 3, 3′, 5, 5′-tetramethylbenzidine (TMB) in DMSO as a chromogen [56]. This test measures hydrogen peroxide (H_2_O_2_) concentration in an antioxidative environment because TMB could react simultaneously with H_2_O_2_ (reaction catalyzed with peroxidase enzyme) and reductive substances such as uric acid (chemical, non-catalyzed reaction). The enzymatic reaction causes TMB oxidation to blue products and chemical reduction to non-colored products. The net reaction is the difference between two opposite oxido-reductive processes at the same substrate. Reaction calibration was performed with the mixture of H_2_O_2_ and uric acid in different ratios, defined from 0 to 100%.

#### 4.2.2. AOPP Concentration Determination

AOPP concentrations were determined using 20 mM phosphate buffer pH 7.4 in reaction with glacial acetic acid and 1.16 M potassium-iodide. The formed complex had an absorbance at a maximum of 340 nm. This reaction was calibrated with chloramine T as a standard, with a concentration range from 10 to 100 µmol/L [57].

#### 4.2.3. MDA Concentration Determination

MDA concentrations were determined using the thiobarbituric acid-reactive substances assay employing the molar absorption coefficient of 1.56 × 10^5^ M^−1^cm^−1^ and spectrophotometry at 535 nm [58]. Thiobarbituric acid reagent consisted of 15% trichloroacetic acid, 0.375% thiobarbituric acid, and 0.25 M hydrochloric acid (HCl).

#### 4.2.4. O_2_^•–^ Level Determination

O_2_^•–^ levels were determined as a rate of nitroblue tetrazolium reduction, as previously described by Auclair and Voisin [59].

#### 4.2.5. SOD Concentration Determination

Plasma SOD concentration was determined using a modified method by Misra and Fridovich [60]. This test relies on the ability of the SOD enzyme to inhibit the autooxidation of epinephrine in an alkaline medium. The maximum absorbance of the pink-colored oxidized product was 480 nm. Epinephrine concentration should be adjusted at a concentration that enables an absorbance change of 0.025 units per minute because this gives a chance for the highest inhibition of spontaneous epinephrine autooxidation. The buffer which gave an alkaline medium was bicarbonate buffer 0.05 mmol/L and pH 10.2. SOD activity was calculated as the percent of inhibition of epinephrine autooxidation.

#### 4.2.6. tSHG Level Determination

tSHG levels were determined with Ellman’s method [61] using 10 mM dinitrodithiobenzoic acid (DTNB) as a reagent. DTNB reacts with aliphatic thiol compounds in a base environment (pH 9.0), generating 1 M p-nitrophenol anion per mole of thiol. Absorbance was measured at 412 nm. Calibration of the method was achieved with the reduced glutathione in a concentration range from 0.1 to 1.0 mM.

### 4.3. Redox Score Calculations

Redox scores were calculated using Z score statistics. We calculated the prooxidative score from all prooxidants and products of its activity measured in this study (PAB, AOPP, MDA, and O_2_^•–^), while the antioxidative score was calculated from all measured antioxidants (SOD and tSHG). Z score was calculated for every parameter using means and standard deviations from the healthy control group according to the formula (Xi-Mean)/SD, where Xi is the individual value of every parameter for every patient. The prooxidative score represents an average value of calculated Z scores of PAB, AOPP, MDA, and O_2_^•–^, and the antioxidative score is an average value of calculated Z scores of SOD and tSHG. OXY-SCORE was calculated as a difference between the prooxidative and antioxidative scores [42,43].

### 4.4. SLFN11 and PD-L1 Level Determination

The SLFN11 and PD-1 protein levels were measured for 97 patients.

The SLFNL11 protein levels in serum were determined using the “sandwich” enzyme-linked immunosorbent assay (ELISA) technique (Wuhan Fine Biotech, Wuhan, China). The test range of detection was from 78 to 5000 pg/mL, with a sensitivity of 46.8 pg/L. The precision of the test was expressed through the intra-assay CV (<8%) and inter-assay (CV < 10%).

The PD-L1 (B7-H1/CD274) protein levels in plasma were determined using the DuoSet ELISA system (R & D Systems Europe Ltd., Abingdon, VA, USA). This system utilizes a “sandwich” ELISA designed for the human B7-H1 protein. The detection range achieved with this method was from 2.0 to 1250 ng/L. As per the manufacturer’s provided data, the reference values for healthy individuals in heparin-plasma samples ranged from 33 to 110 ng/L.

### 4.5. Statistical Analysis

Statistical analysis was performed using IBM SPSS Statistics v20 software (IBM Corporation, Armonk, NY, USA). The Kolmogorov–Smirnov test and the Shapiro–Wilk test were used to test the normality of data. We utilized independent samples *t*-test and analysis of variance (ANOVA) with Tukey’s Hinges post hoc tests to test the differences between the groups of continuous data in a normal distribution. Related samples Wilcoxon signed-rank test was used for matched samples, and independent samples Mann–Whitney U and the Kruskal–Wallis tests were used for independent samples to test the differences between the groups of continuous data not normally distributed. Univariate and multivariate logistic regression analyses were applied with survival status (one year and three year) as dependent variables and Kaplan–Meier survival analysis. A factorial analysis (principal component analysis (PCA)) was performed to reduce the number of variables into a smaller number of factors formed from the parameters with the same level of variability. The Kaiser–Meier–Olkin test was used for the analysis of sampling adequacy. For sphericity existence, we used Bartlett’s test. *p*-values less than or equal to 0.05 were considered significant.

## 5. Conclusions

In conclusion, our study showed that OXY-SCORE can be used as a potential prognostic biomarker for overall survival in CRC patients. SLFN11 protein concentrations measured in serum using the ELISA method also harbor prognostic potential for one-year and three-year survival. Additionally, only histopathology-analysis-related factors showed prognostic value in univariate and multivariate binary logistic regression analyses of the PCA-extracted factors. Since this is, to our best knowledge, the first study to evaluate OXY-SCORE values and SLFN11 concentrations in the serum of CRC patients, further studies should validate these findings in an independent patient cohort. Our future investigations will focus on SLFN11 gene expression and the evaluation of mRNA levels in relation to serum concentrations obtained in the present study.

## Figures and Tables

**Figure 1 ijms-24-15083-f001:**
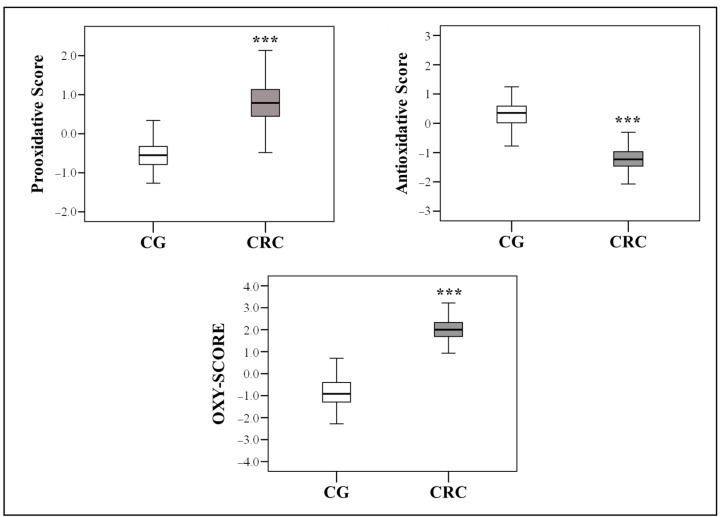
Differences in prooxidative scores, antioxidative scores, and summary OXY-SCOREs between CRC patients and the healthy control group. The box defines the interquartile range (25–75th percentile), the line within the box is the median value, and the whiskers’ ends represent the group’s minimum and maximum values. *** *p* < 0.001 vs. control group using the Mann–Whitney U test. Abbreviations: CG—control group; CRC—colorectal cancer patients.

**Figure 2 ijms-24-15083-f002:**
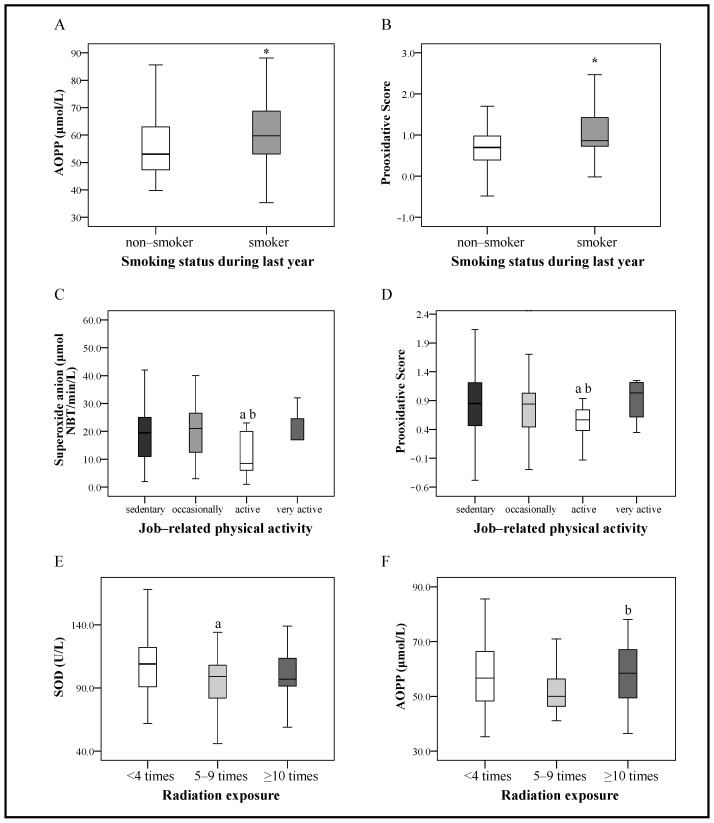
The influence of smoking status during the last year, job-related physical activity, and radiation exposure on measured and calculated parameters of redox status in CRC patients presented as box plots. The box defines the interquartile range (25–75th percentile), the line within the box is the median value, and the whiskers’ ends represent the group’s minimum and maximum values. (**A**) The influence of smoking status during last year on AOPP concentrations. (**B**) The influence of smoking status during last year on prooxidative score values. * *p* < 0.05 vs. non-smokers by Mann–Whitney U test. (**C**) The influence of job-related physical activity on superoxide anion levels. (**D**) The influence of job-related physical activity on prooxidative score values. ^a,b^
*p* < 0.05 vs. sedentary and occasional activity, respectively. (**E**) The influence of radiation exposure on SOD concentrations. (**F**) The influence of radiation exposure on AOPP concentrations. ^a,b^
*p* < 0.05 vs. <4 times and 5–9 times, respectively, using the Mann–Whitney U test.

**Figure 3 ijms-24-15083-f003:**
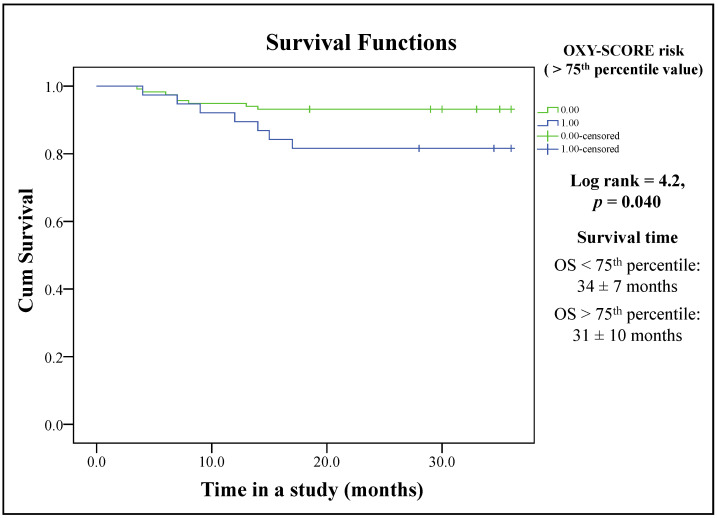
The Kaplan–Meier analysis of the CRC patients’ survival according to OXY-SCORE value risk (>75th OS percentile value). Abbreviations: OS—overall survival.

**Figure 4 ijms-24-15083-f004:**
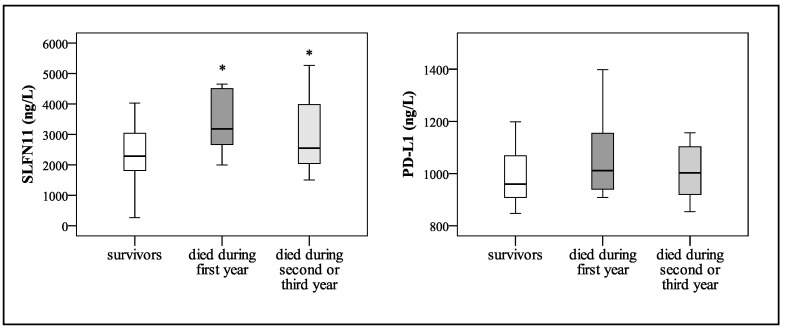
SLFN11 and PD-L1 protein serum concentrations in CRC patients according to survival status during the three years of study. The box defines the interquartile range (25–75th percentile), the line within the box is the median value, and the whiskers’ ends represent the group’s minimum and maximum values. * *p* ≤ 0.05 vs. survivors (Mann–Whitney U test).

**Table 1 ijms-24-15083-t001:** The socio-demographic and clinical data of the CRC patients.

Parameter	N (%)/Mean ± SD
Number of patients (N)	155
Age	69.2 ± 10.0
Gender	
Male	100 (64.5)
Female	55 (35.5)
Smoking status	
Never	63 (40.6)
Smoker	36 (23.2)
Ex-smoker	56 (36.2)
Alcohol use	
No	80 (51.7)
Yes	65 (41.9)
N/A	10 (6.4)
Education level	
Elementary	15 (9.7)
High school	73 (47.1)
University	57 (36.8)
N/A	10 (6.4)
Radiation exposition	
Never	2 (1.3)
1–4 times	61 (39.3)
5–9 times	46 (29.7)
>10 times	34 (21.9)
N/A	12 (7.8)
Physical activity (job related activity)	
Sedentary (office)	77 (49.7)
Rare physical activity	44 (28.4)
High physical activity	18 (11.6)
Very high physical activity	4 (2.6)
N/A	12 (7.7)
Histological type of tumor	
Adenocarcinoma	129 (83)
Mucinous adenocarcinoma	26 (17)
T stage	
T1	7 (4.5)
T2	18 (11.6)
T3	103 (66.5)
T4	27 (17.4)
N stage	
N0	78 (50.3)
N1	53 (34.2)
N2	24 (15.5)
Lymphovascular (LV) invasion	
No	65 (42)
Yes	86 (55.5)
LVx	4 (2.5)
Perineural invasion	
No	133 (85.8)
Yes	22 (14.2)
Primary tumor histopathological stage	
I	16 (10.3)
II	62 (40)
III	69 (44.5)
IV	8 (5.2)
Dukes staging	
A	18 (11.6)
B	61 (39.3)
C	67 (43.2)
D	9 (5.9)
Astler-Coller staging	
A	8 (5.2)
B1	11 (7.1)
B2	59 (38.1)
B3	2 (1.3)
C1	6 (3.9)
C2	57 (36.8)
C3	5 (3.2)
D	9 (4.4)
Residual status	
R0	140 (90.3)
R1	13 (8.4)
Rx	2 (1.3)

Abbreviations: SD—standard deviation; N/A—not available; LVx—lymphovascular invasion cannot be determined; Rx—R status cannot be determined.

**Table 2 ijms-24-15083-t002:** Redox status parameters in CRC patients and the healthy control group. Values are presented as median (range).

Parameter	CRC Patients	Control Group	*p*
PAB (U/L)	64 (54–76)	68 (55–93)	ns
AOPP (μmol/L)	54.7 (47.6–65.0)	18.8 (17.8–20.8)	<0.001
MDA (μmol/L)	2.96 (2.52–3.63)	2.67 (2.26–3.04)	0.002
O_2_^•–^ (μmol NBT/min/L)	19 (10–25)	11 (2.5–16.5)	<0.001
SOD (U/L)	103 (89–114)	132 (123–137)	<0.001
tSHG (mmol/L)	0.294 (0.241–0.373)	0.360 (0.322–0.391)	0.002

Abbreviations: ns—non-significant.

**Table 3 ijms-24-15083-t003:** PCA-extracted factors from clinical and redox status parameters in a group of CRC patients.

Factor	Variables with Loadings	Factors’ Percent of Total Variability
Histopathology-analysis-related factor	Astler-Coller staging 0.975 Pathohistological staging 0.958 Regional lymph nodes infiltration 0.891	34%
Immuno-modulatory-oxidative stress-related factor	SLFN11 (ng/L) 0.720 AOPP (μmol/L) –0.612 PD-L1 (ng/L) 0.553	19%
Redox-status-related factor	tSHG (mmol/L) 0.794 O_2_^•–^ (μmol NBT/min/L) 0.728	16%

**Table 4 ijms-24-15083-t004:** Univariant and multivariant binary logistic regression analyses of PCA-extracted factors for one-year mortality prediction.

PCA Factors	Univariant Analysis				Multivariant Analysis			
	B (SE)	Wald Coefficient	OR (95% CI)	*p*	B (SE)	Wald Coefficient	OR (95% CI)	*p*
Histopathology-analysis-related factor	1.085 (0.419)	6.69	2.9 (1.3–6.7)	0.010	1.094 (0.420)	6.77	3.0 (1.3–6.8)	0.009
Immuno-modulatory-oxidative stress-related factor	−0.166 (0.342)	0.237	0.8 (0.4–1.6)	0.627	−0.092 (0.336)	0.074	0.9 (0.5–1.8)	0.785
Redox-status-related factor	−0.186 (0.380)	0.240	0.8 (0.4–1.7)	0.624	−0.304 (0.426)	0.508	0.7 (0.3–1.7)	0.476

Abbreviations: B—unstandardized regression weight; SE—variation in unstandardized regression weight; OR—odds ratio; CI—confidence interval.

## Data Availability

The data used during this study are available from the corresponding author upon reasonable request.

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
