# Peer review of "Analysis of the Prognostic Potential of Schlafen 11, Programmed Death Ligand 1, and Redox Status in Colorectal Cancer Patients"

_ijms, 2023, doi:10.3390/ijms242015083_

Round 1

Reviewer 1 Report

Manuscript: 2575273

In this manuscript titled "Analysis of the prognostic potential of SLFN11, PD-L1, and redox status in colorectal cancer patients" by Miladinov et al., the authors aim to study the prognostic potential of three different serum parameters including redox potential, SLFN11 and PD-L1 levels in colorectal cancer patients. Though the study involves statistically satisfying number of patients, but the design doesn’t have any potential rationale. This study can gain by addressing the following concerns.  

I would rather suggest authors to provide rationale for selecting three totally unconnected biomarkers to study their prognostic effect. Authors can emphasize the whole study on redox parameters or provide a connection with redox status of the patients with SLNF11 and PD-L1 proteins. I would nice if the study is focused on any particular biological aspect rather than being too much deviated.

In discussion, authors acknowledged the discrepancy in the results they observed with a previous study. “Conversely, in a in immunohistochemical analysis by Deng et al., patients with higher SLFN11 expression had significantly longer three-year survival [45]. A possible explanation for this discrepancy could be that the tumor burden is higher in people with shorter survival; higher concentrations of SLFN11 are released into the blood, and the measured concentrations in the serum are higher. Rather than providing possible explanation, I would suggest authors to perform immunohistochemical analysis of the tumors and provide explanation on SLFN11 ration between tumor tissue and serum of patients.

SLFN11 is known to vary with the mutational status. Do authors have mutational information on these patients?

All the methodologies used to measure redox parameters were developed in 19th century based on the citations provide by authors. I would recommend authors to include positive control in every experiment to check the fidelity of the assay. Example, AOPP Assay Kit from Abcam has a positive control AOPP Human Serum Albumin conjugate.

Table 2: Looking at median and range of MDA between CRC patients and control group, p value of 0.002 is questionable. I recommend authors to revisit this data and double check.

Authors claims that redox status disturbance was measured through AOPP, and smoking caused significant difference in AOPP and prooxidative score. Was prooxidative score dependent only on AOPP here? Why only AOPP, why authors did not include other redox parameters?

In Kaplan Meier analysis authors used 135 CRC subjects at risk. How many patients were distributed above or below 75th percentile? Please provide subjects at risk for each group and hazard ratio.

SOD Levels and activity correlation is needed. Because there are isoforms of SODs present just measuring SOD activity will not provide much information on the redox status.

Why have authors not measured glutathione reduced and oxidized levels in the serum?

Please provide proper methodology since the references cited are very old and the question will be: Are these methods going to translate from bench to bedside to evaluate the prognosis of CRC patients. Although Pro-oxidant antioxidant balance oxidative is a key factor in many diseases, the balance of oxidant-antioxidant as a clinical laboratory method is not routinely measured, and its main reason is the lack of an accepted method in the world .

Author Response

Dear Madam/Sir,

Thank you for your valuable feedback and helpful comments and suggestions. Please find attached a point-by-point reply to your remarks. The changes have been made in the revised manuscript accordingly.

Best regards,

Prof. dr. Goran Barisic

Reviewer 2 Report

The abstract provides a succinct overview of a study exploring the roles of Schlafen 11 (SLFN11) protein, Programmed death ligand 1 (PD-L1) protein, and redox status parameters as potential prognostic biomarkers in colorectal cancer (CRC) patients. The abstract is well-structured, highlighting the significance of the investigated proteins and parameters in the context of DNA damage, immune evasion, and cellular homeostasis.  Only two things need to be addressed:

1. It would be better to separate the conclusion and discussion. 

2. The author should add SD value for every data.

Moderate editing of English language required

Author Response

(The authors gave the same response as above.)

Round 2

Reviewer 1 Report

Authors have tried to address most of the concerns yet stated funding as limiting factor for most of the queries raised. However, authors need to address at least the following concerns.

“The above discussion on the rationale for studying oxidative stress, SLFN11, and PD-L1 was added as the second paragraph of the Discussion section on page 11, lines 227-234”

Please move this section to introduction. Rationale for the study must be always stated upfront as in introduction.

“We are not able to perform immunohistochemical analysis for this patient’s cohort since tumor tissues are not available for all patients anymore.”

Authors can still perform Immunohistochemistry on available patient tumor tissues and compare it with serum levels. And to justify the number of patient tumors add a sentence as one of the limitation on the availability of patient tumors.

“The prooxidative score was only significant among calculated parameters, while AOPP was only significant among measured parameters influenced by smoking, so only these two results were presented”.

Please include this in discussion

“Percentiles are measures that divide a whole group of numbers at 100 parts, so the 75th percentile divides a group of numbers at ¾ below the cut-off value and ¼ above the cut-off value. The exact numbers are 101 patients below the cut-off OXY-SCORE value and 34 patients above this value”.

This uneven distribution of patients above or below the cut-off doesn’t reflects the actual survival. However, owing to the limitation of the number of patients authors should discuss this in discussion that the number of patients above the cut-off value were less and needs more investigation in future to predict the survival outcome.

Author Response

Dear Madam/Sir,

Thank you again for your valuable feedback and helpful comments and suggestions. Please find attached a point-by-point reply to your additional remarks. The changes have been made in the revised manuscript accordingly.

Best regards,

Prof. dr. Goran Barisic

Round 3

Reviewer 1 Report

Authors have addressed and revised the manuscripts as per the concerns raised.